# Highly stretchable and transparent ionic conducting elastomers

Lei Shi[1], Tianxiang Zhu[1], Guoxin Gao[1], Xinyu Zhang[2], Wei Wei [1], Wenfeng Liu[1] & Shujiang Ding[1]

Traditional elastomers are mostly dielectrics; existing conductive elastomers are conductive composites with electric conductors. Herein, we introduce a series of ionic conducting elastomers (ICE) by salt in polymer strategy. The ICEs possess good stretchability, transparency and ionic conductivity. Moreover, the ICEs exhibit very high stability in air, under high temperature and voltage, with excellent adhesion properties and no corrosive effects to metal electrodes. Touch sensors are fabricated using these ICEs—impedance spectra and impedance complex plane are tested and analyzed to clarify different stimulus of the touch sensors. These ICEs provide possibilities for flexible electronics and soft machines.

[1] Department of Applied Chemistry, School of Science, MOE Key Laboratory for Nonequilibrium Synthesis and Modulation of Condensed Matter, State Key Laboratory of Electrical Insulation and Power Equipment, Xi'an Jiaotong University, Xi'an 710049, People's Republic of China. [2] Department of Chemical Engineering, Auburn University, Auburn, AL 36849, USA. Correspondence and requests for materials should be addressed to S.D. (email: dingsj@mail.xjtu.edu.cn)

Elastomers including natural or synthetic rubbers have been studied and utilized for centuries and enabled diverse modern technologies, benefiting from their soft and deformable properties and diverse functionality[1–4]. Function elastomers with conductive properties provide tremendous opportunities for flexible electronics and soft machines, which have pushed for a rapid growth in the related research fields these years[5–7]. They can serve as stretchable electrodes and wires, components in capacitive or resistive circuit elements, such as sensors, actuators, and cables. Existing conductive elastomers are mostly conductive composites: rigid conductors including metals, carbons, and conducting polymers are filled into or coated on elastomers to obtain conductivity and elasticity[8–13]; conductive fluids such as liquid metals[14–16] and liquid electrolytes[17,18] are injected into pipelines of elastomer substrates to implement functionality. Traditional conductive composite elastomers are hard to achieve transparency and elasticity at the same time, which hinders their application in optical related soft electronics. In addition, the manufacturing processes are complex and costly.

Herein, we intend to introduce a concept of conductive elastomers, ionic conducting elastomers (ICE), achieving functionalities which are extremely difficult or even impossible to realize by traditional conductive composite elastomers. Distinguished from electronic conductors, ionic conductors or so called electrolytes use ions as current carriers. By using ionic conductors, a series of devices have been reported[19–24]. For example, Sun group have demonstrated a highly stretchable and transparent touch panel by using hydrogel electrolytes. The panel can be operated under large areal strain with 98% transmittance for visible light. Such performances are hardly achieved by traditional electronic conductors[20].

ICE shows good stretchability, transparency, and conductivity at the same time. ICE is synthesized by salt in polymer strategy, achieving conductivity by ionic transportation through polymer chains. On the other hand, since the material is solvent free, it is quite stable in air, shows neither weight loss nor the decay of its stretchability, transparency, and conductivity. Additionally, ICE possesses great high-temperature stability, with a decomposition temperature up to 300 °C, which makes it possible to use ICE at high temperature. Moreover, ICE is not corrosive to ordinary metal electrodes, achieving long-term stability of ICE-metal joints, because it does not contain water and absorb moisture form air (the polymer matrix is hydrophobic). And, its decomposition voltage is several times higher than hydrogels. Therefore, higher voltages can be applied to the ICEs. With such advantages, ICE may be an ideal material for engineering ionic devices. We have demonstrated its application for a touch sensor.

## Results

**Preparation of ICE.** ICE was fabricated by instant photocuring process. Lithium bis(trifluoromethane sulfonimide) (LiTFSI) was employed as the electrolyte salt, and butyl acrylate (BA), polyethyleneglycol diacrylate (PEGDA), 1-hydroxycyclohexyl phenyl ketone (photo-initiator 184) were used as monomer, crosslinker, and photo-initiator, respectively (Fig. 1a). The facile fabrication process of ICE includes a few steps: firstly, LiTFSI powder, PEGDA and photo-initiator 184 were dissolved in BA liquid to form a transparent precursor solution. The molar percentage of PEGDA and photo-initiator 184 to BA were 0.1 and 1% throughout the entire experiments, respectively. The molar concentration of LiTFSI was fixed at 0.5 M. Then, the solution was injected into a release film coated glass mold. ICE was cured in 10 minutes by ultraviolet light irradiation (365 nm, 400 W power). The choice of using LiTFSI as electrolyte salt was the critical point of this system. As a frequently-used electrolyte salt for lithium ion batteries[25,26], LiTFSI easily dissolves in varies of solvents and polymers and form electrolytes with high conductivity and transparency. After the success of LiTFSI in the salt in polymer system, several other salts, such as LiClO₄ (lithium perchlorate) and LiCl (lithium chloride) were tested, but failed to obtain transparent and conducting elastomers using these salts, due to separation during photocuring process.

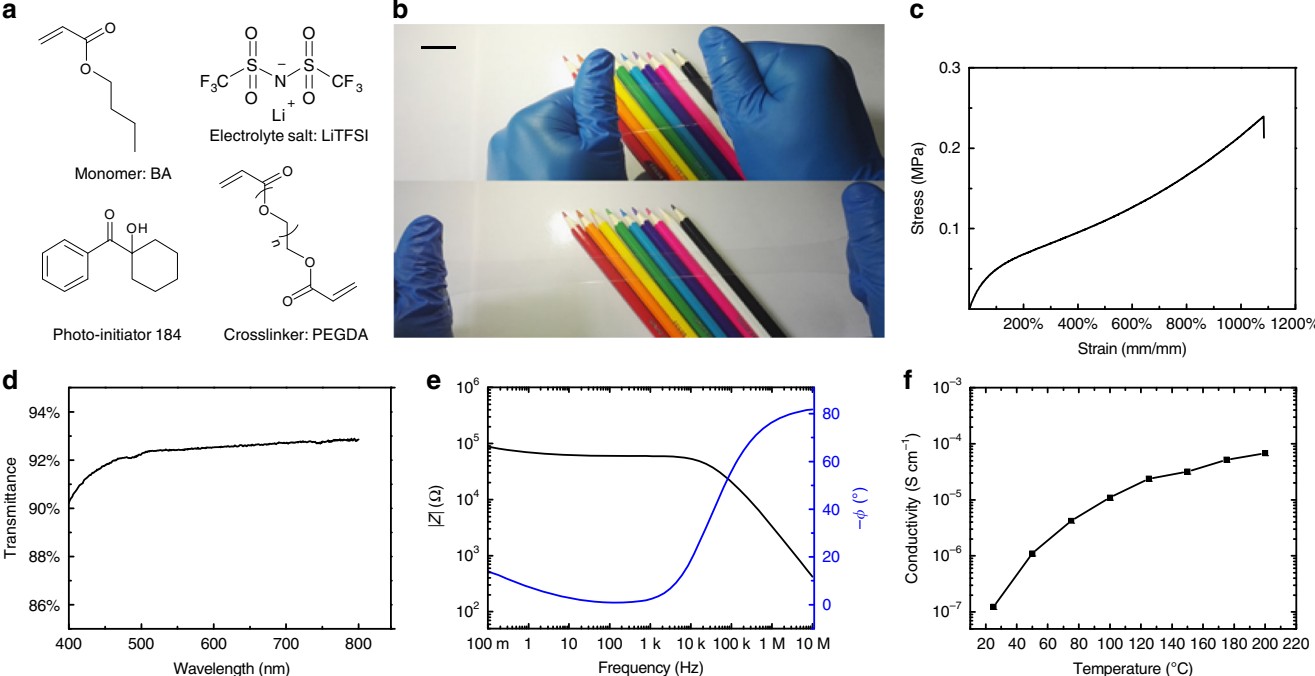

**Fig. 1** Synthesis and properties of ICE. **a** Molecular structures of ICE precursors. **b** Photographs of as-prepared ICE, demonstrating the good stretchability and high transparency of the ICE (scale bar: 2 cm). **c** Stress-strain curves of ICE. **d** Transmittance test of ICE in the visible range. **e** Plot of impedance magnitude (|Z|, black line) and negative phase angle (−ϕ, blue line) versus testing frequency. **f** Plot of conductivity versus testing temperature

**Basic properties of ICE**. Photographs in Fig. 1b demonstrated the good transparency and stretchability of ICE. Tensile tests were carried out using ICE dumbbell samples with dimensions of $12.0 \times 2.0 \times 2.0$ mm$^3$. The tensile tests were performed on an electronic tensile machine with a 50-N load cell, and the stretching rate was set at 100 mm min$^{-1}$. As shown in Fig. 1c, ICE possesses good stretchability, with elongation at break of ~1100%. The transmittance of ICE was measured using an UV-Vis spectrophotometer. As shown in Fig. 1d, ICE was fully transparent in visible light region, a 1-mm thick sample showed a transmittance of 92.4% at 550 nm. We measured the electrical properties by an alternating-current impedance spectroscopy. A 1-mm thick sample was sandwiched between two copper electrodes with diameter of 30 mm. Similar to liquid electrolytes and gel electrolytes, ICE showed impedance frequency dependency (Fig. 1e). The impedance magnitude ($|Z|$) sharply decreased and span several orders of magnitude at high frequency range (10 KHz–10 MHz), while at low frequency range (100 mHz–10 KHz), $|Z|$ remained in the ballpark of the starting point. On the other hand, the negative phase angle ($-\phi$) nearly maintained as constant at low frequency range; when the frequency increased from 1 KHz to 10 MHz, $-\phi$ significantly changed from ~0° to ~80°. The capacitance ($Cp'$) versus frequency was plotted in Supplementary Fig. 1. At low frequency range, as the testing frequency decreased, the $Cp'$ increased sharply and span several orders of magnitude, from $10^{-11}$ F (10 KHz) to $10^{-6}$ F (100 mHz). While at high frequency range, $Cp'$ maintained constant at $10^{-11}$ F. Obviously, the testing sandwich contained capacitance and resistance characteristics at the same time. These phenomena were quite different from electric conductors. We measured impedance spectroscopy of a carbon sponge (Supplementary Fig. 2), its $|Z|$ value maintained as constant through the whole testing frequency range (100 mHz–10 MHz), so did the $\phi$ (~0°). Thus, when employing ionic conductors, the applying frequency needs to be well considered. We read the bulk resistance (114 KΩ) of ICE from the Nyquist plot of impedance spectrum (Supplementary Fig. 3)[27,28]. Its conductivity was about $1.27 \times 10^{-7}$ S cm$^{-1}$ at room temperature (20 °C), calculated by the equation $\sigma = L/SR$, where $L$ corresponded to the thickness of ICE, $S$ corresponded to the effective overlap area, and $R$ corresponded to the bulk resistance. ICE exhibited higher conductivity as the temperature increased, as shown in Fig. 1f, the conductivity of ICE was $6.80 \times 10^{-5}$ S cm$^{-1}$ at 200 °C. This can be explained as follows: higher temperature contributed to more intense movement of polymer chains and ions, resulting in easier ion transport and higher conductivity.

**Characteristics of ICE**. Because ICE was solvent free, it possessed many special characteristics, which were different from traditional gel electrolytes. Figure 2a was the thermogravimetric curve of ICE, it showed ICE possesses ultra-high decomposition temperature (up to 335 °C in N$_2$ and 320 °C in air), which made the operating temperature can even exceed 100 °C. We compared the stability of ICE and hydrogel electrolyte at different constant temperatures (25, 50, and 100 °C) in open air. The hydrogel for comparison experiments was polyacrylamide (PAAm) hydrogel containing 2 M lithium chloride (LiCl) salts. The hydrogel was synthesized via the method in Ref. 20. Hydrogel electrolytes (Supplementary Fig. 4) dehydrated instantly at high temperature (50 and 100 °C), with weight loss of about 70% in just a few minutes. Even at 25 °C, it lost half of its weight in several hours. Finally, its characteristics such as, shape, mechanical properties, conductivity, and transparency were extremely changed. While ICE was quite stable at these temperatures, its weight remained unchanged, as shown in Fig. 2b.

Practically, ionic conductors have to be connected with metals to implement functions, regardless of receiving electrical stimuli or output electric signals. The stability of electrolyte–metal junctions has to be concerned. Photographs in Fig. 2c were ICE-metal and hydrogel–metal junctions after placing in open air and at room temperature for one week. Hydrogel electrolyte dehydrated, deformed and showed corroding effect on metals (aluminum, copper, tin, and steel), while ICE was quite stable and no corrosion to metals, because ICE do not contain water and it does not absorb moisture from air. On the other hand, ICE can attach to metals strongly, photographs in Fig. 2d demonstrated adhesive property of ICE. Simply by attaching ICE to surfaces of different materials, junctions formed immediately, dispensed with surface modifications. Supplementary Movie 1 showed the attaching process and the stability of the junctions. The ICE–metal junctions and ICE–ICE junction were stable even at stretch. This characteristic made ICE excellent candidate for fabrications of ionic circuits. The strong adhesion property of the elastomer originated from the polymer substrate polybutylacrylate (PBA), a commonly used Acrylate Adhesive. By 90°-peeling tests, the measured peeling forces per width of ICE on various substrates (Glass, copper, aluminum and ICE) were shown in Fig. 2e. The measured interfacial toughness of the hybrid samples were around 150 J m$^{-2}$. And, decomposition voltage of ICE was several times higher than that of hydrogel electrolytes, as shown in Fig. 2f. The decomposition voltage of ICE and hydrogel electrolyte were tested via Linear-Sweep-Voltammetry, the scan rate was set at 0.5 mV s$^{-1}$ at the range from 0 to 10 V. Detected current of hydrogel electrolyte increased sharply as the applying voltage exceeded ~1 V, accompanied by bubbles generating at the interfaces of electrodes and electrolyte, indicating the decomposition of water by electrolysis. ICE was quite stable through the whole voltage range, indicating higher voltages can be applied on ICE electrode.

**Copolymerization induced conductivity improvement**. The concept of ICE can be extended beyond neat PBA system. Figure 3a showed several kinds of monomers that can be copolymerized with BA. In this study, copolymer ICEs were synthesized by the same photocuring process as PBA-based ICE, with the volume ratio of comonomers to BA at 1:1. Comonomers include 2(2-ethoxyethoxy) ethyl acrylate (EOEOEA), Poly(ethylene glycol) methyl ether acrylate (MPEGA), and vinylene carbonate (VC). MPEGA with different molecular weights were employed, MPEG350A and MPEG550A, 350 and 550 represented the molecular weight of PEG chain. These comonomers greatly enhanced the conductivity. As shown in Fig. 3b, c, the bulk resistance copolymer ICEs were greatly decreased, the conductivity of copolymer ICEs reached up to $10^{-6}$–$10^{-5}$ S cm$^{-1}$. The comonomers were frequently employed for solid-state polymer electrolytes and they enhanced the conductivity. As an example, by introducing PMPEG550A into the polymer substrate helped increasing ionic conductivity. The homopolymer of PMPEG550A with 0.5 M LiTFSI have a ionic conductivity of $6.3 \times 10^{-5}$ S cm$^{-1}$. We showed that as the volume percentage of MPEG550A increases, the conductivity of the copolymer increases (Supplementary Fig. 5, 6). While PMPEG550A solid electrolyte was brittle and unstretchable, in this study, copolymerization achieves high conductivity and stretchability.

**ICE-based touch sensors able to identify touch and stretch**. Touch sensors were fabricated using copolymer ICE (co-MPEG550A, conductivity of $1.93 \times 10^{-5}$ S cm$^{-1}$ at room temperature). We illustrated impedance spectra detecting signals from different types of stimuli. Figure 4a was the schematic diagram

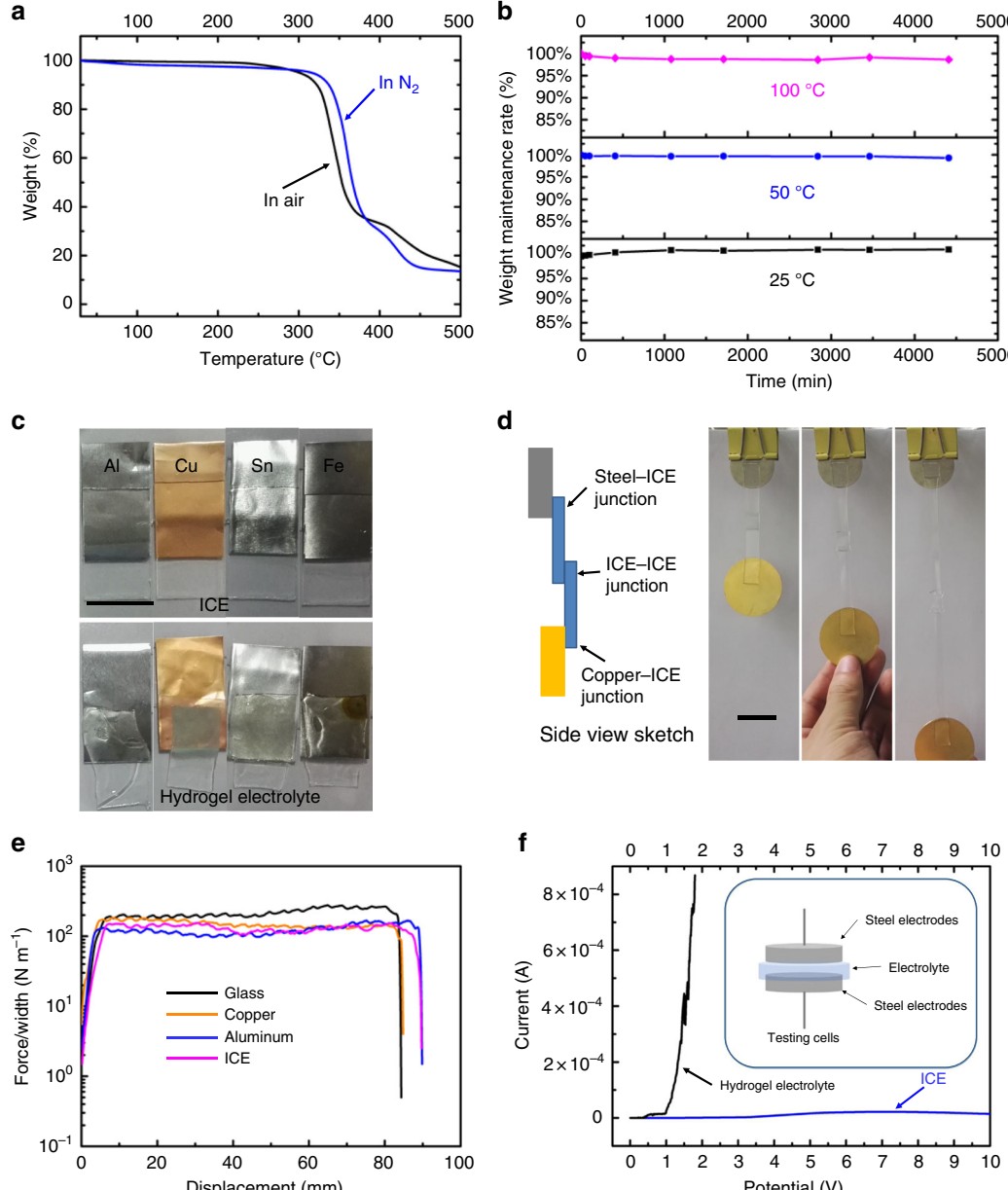

**Fig. 2** Characteristics of ICE. **a** Thermogravimetric curve of ICE (blue line, in N₂ and black line, in air). **b** Environmental stability testing of ICE at open air. Plots of weight maintenances of ICE at different temperature versus time. Black, blue, and magenta lines, represent the testing temperature of 25, 50, and 100 °C, respectively. **c** Photographs of ICE–metal and hydrogel–metal junctions after placing at open air and room temperature for 1 week. ICE was quite stable and not corrosive to metals, including aluminum (Al), copper (Cu), tin (Sn), and steel (Fe), while hydrogel electrolyte dehydrated, deformed and showed corroding effect on metals. The hydrogel for comparison experiments was polyacrylamide (PAAm) hydrogel containing 2 M lithium chloride (LiCl) salts. (scale bar: 2 cm) **d** Photographs demonstrating adhesive property of ICE. The junctions were stable even at stretch. (scale bar: 2 cm) **e** The measured peeling forces per width of ICE on various substrates by 90°-peeling tests. A stiff backing was introduced to prevent elongation of ICE while peeling. Black, orange, blue, and magenta lines represent the testing substrates of glass, copper, aluminum, and ICE, respectively. **f** Decomposition voltage testing of ICE (blue line) and hydrogel electrolyte (black line) via Linear Sweep Voltammetry (LSV). Inset at the upper right corner is the sketch of testing cell

of an uncovered touch sensor with four different states, original, stretched, touched, stretched, and touched. As shown in Fig. 4b, c, when stretched, the |Z| of the sensor increased and the −φ (negative phase angle) changed slightly (Supplementary Movie 2); when touched, the |Z| and -φ versus frequency curves showed great difference from original or stretched states, i.e., the |Z| exhibits a large peak in the frequency range of 1 KHz–1 MHz, the −φ changed so much from positive value (~80°) to negative value (~−40°) in the same frequency range. Figure 4d was the Nyquist plots of the impedance spectra, differences were showed in the complex plane.

Obviously, at low frequency range (tens of KHz to 1 KHz), touched sensor exhibited inductance characteristics, because, φ became positive values. Since human can be regarded as an ionic conductor, when finger touched the sensor, human became part of the circuit. Finger conducts currents and introduced new elements into the circuit, resulting in the change of the circuit characteristics. We set a single frequency (f = 20 KHz) to detect signals from different stimulus, as shown in Fig. 4e, different stimulus appeared at different regions in impedance complex plane, which means the sensor is very selective to different stimuli.

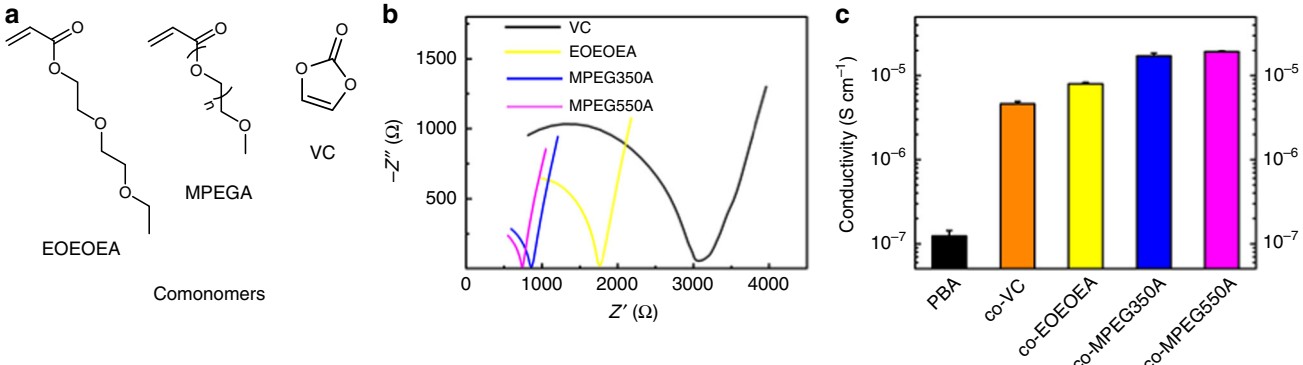

**Fig. 3** Enhancing the conductivity of ICE by copolymerization. **a** Molecular structures of comonomers used for enhancing the conductivity. **b** Nyquist plots of impedance spectra of different copolymer ICEs. Black, yellow, blue, and magenta lines represent the copolymer monomers of VC, EOEOEA, MPEG350A, and MPEG550A, respectively. Intersections of the curves with the real axis correspond to resistances of testing samples. **c** Histogram of conductivities of PBA (original ICE, black bar) and different copolymer ICEs (VC, orange bar; EOEOEA, yellow bar; MPEG350A, blue bar; MPEG550A, magenta bar). Error bars in the figure represent standard error of the mean of the data ($n = 3$)

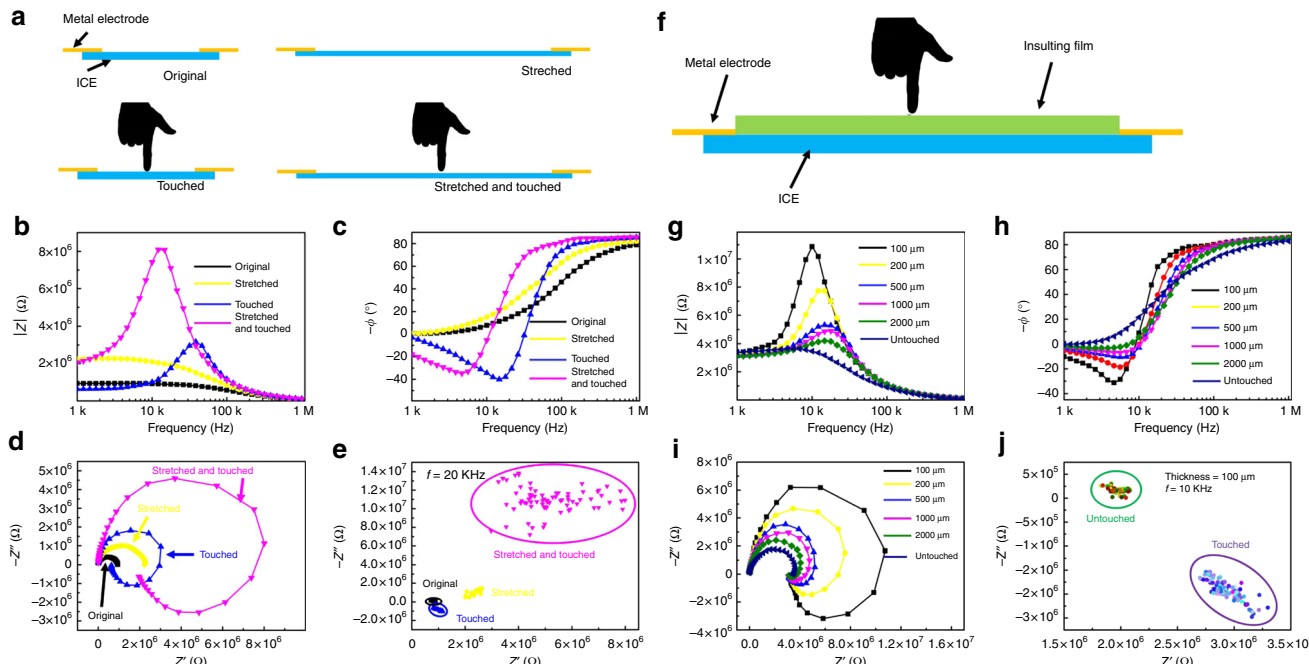

**Fig. 4** Impedance spectra of ICE based touch sensors. **a** Schematic diagram of uncovered touch sensor at different states. Plots of impedance magnitude ($|Z|$) and negative phase angle ($-\phi$) versus testing frequency of uncovered touch sensor (**b, c**) at different states (original, black line; stretched, yellow line; touched, blue line; stretched and touched, magenta line). **d** Nyquist plots of impedance spectra of uncovered touch sensor at different states. **e** Repeatedly detected data of uncovered touch sensor at a single frequency ($f = 20$ KHz) in impedance complex plane (original, black dots; stretched, yellow dots; touched, blue dots; stretched and touched, magenta dots). **f** Schematic diagram of insulating film covered touch sensor. Plots of impedance magnitude ($|Z|$) and negative phase angle ($-\phi$) versus testing frequency of insulating film covered touch sensor (**g, h**), varies thickness of insulating film were tested (100 μm, black line; 200 μm, yellow line; 500 μm, blue line; 1000 μm, magenta line; 2000 μm, green line; untouched, deep blue line). **i** Nyquist plots of impedance spectra of insulating film covered touch sensor. **j** Repeatedly detected data of insulating film covered touch sensor at a single frequency ($f = 10$ KHz) in impedance complex plane. Green circle, untouched, purple circle, touched

Interestingly, insulating film covered sensor can still sense finger touch (Supplementary Movie 3). Figure 4f was the schematic diagram of the insulating film covered sensor. Same as the uncovered sensor, finger touch greatly changed the characteristics. Insulating films (silicone rubber) with different thickness were tested. Even with a 2000 μm thickness silicone rubber cover, a finger touch can cause obvious differences of the spectra, as shown in Fig. 4g, h, i. As the thickness decreased, the change of $|Z|$ and $-\phi$ became more significant, thinner cover resulted in easier circuit change when finger touched. We set a

single frequency ($f = 10$ KHz) to detect touch signals, as shown in Fig. 4j, touch stimulus appeared at different region from the original untouched region in impedance complex plane. A fitting circuit was given in Supplementary Fig. 7.

## Discussion

In conclusion, by salt in polymer strategy, we introduced a series of material, ICE. The as prepared ICEs possessed high stabilities, including air stability, high temperature stability, adhesion

stability, non-corrosive stability, and high voltage stability. We demonstrated the use of ICEs making touch sensors and introduced impedance spectra and impedance complex plane to clarify different stimulus of the touch sensors.

## Methods

**Materials**. All the chemicals used in this work are analytical reagents.

**Synthesis of the ICEs**. Firstly, LiTFSI powder, PEGDA, and photo-initiator 184 was dissolved in BA liquid to form a transparent precursor solution. The molar percentage of PEGDA and photo-initiator 184 to BA were 0.1% and 1% throughout the entire experiments, respectively. The molar concentration of LiTFSI was fixed at 0.5 M. Then, the solution was injected into a release film coated glass mold. ICE was cured in 10 min by ultraviolet light irradiation (365 nm, 400 W power). We controlled the thickness of the ICE by adjusting the thickness of the silicone spacer. For copolymer ICEs, the volume ratio of comonomers and BA were set at 1:1, molar percentage of photo-initiator 184 were set at 1%, no use of crosslinkers.

**Synthesis of hydrogel electrolyte for comparison**. The molar concentrations of Acrylamide (AAm) and LiCl aqueous solutions were 2.17 and 2 M. The crosslinker N,N-methylenebisacrylamide (MBAA) 0.06 wt.% and the initiator Ammonium persulfate (APS) 0.16 wt.%, with respect to the AAm monomer, were added. After sonicating and degassing the mixture in a vacuum chamber, N,N,N',N'-tetramethylethylenediamine (TEMED) 0.25 wt.%, with respect to the weight of the AAm monomer, was added as the accelerator. The solutions were poured into glass molds. The gel was prepared after 1 h.

**Mechanical tests**. The mechanical tests were performed on an electronic tensile machine (CMT6503, MTS) with a 50-N load cell. The samples were cut into dumbbell shape (testing measure of $12.0 \times 2.0 \times 2.0$ mm$^3$) for tests. The stretching rate was set at 100 mm min$^{-1}$.

**Transparency tests**. The transparency tests were performed on an UV-Vis spectrophotometer (PE Lambda950, Instrument Analysis Center of Xi'an Jiaotong University). The samples for tests were 1-mm thickness. The wavelength for testing were set from 800 to 400 nm. The reference for measuring transparency was air.

**Impedance tests**. The impedance tests were performed on a broadband dielectric/impedance spectrometer (Novocontrol GmbH). The samples were 1-mm thickness, the testing copper electrodes were diameter of 30 mm, testing $V_{rms}$ (voltage effective value) was set at 1 V. The samples were treated without metal spraying on surfaces. For testing touch sensors, the impedance tests were performed on an electrochemical workstation (CHI660E), testing $V_{rms}$ was set at 0.5 V.

**Thermogravimetric analysis (TGA) measurements**. The TGA measurements were performed on a TGA Q 5000 via scanning a temperature range from room temperature up to 500 °C (10 °C min$^{-1}$) under flowing N$_2$ and air by using alumina crucible.

**Peeling tests**. A 1 mm thick ICE sheet were cut into a shape of 1 mm × 20 mm × 80 mm. A stiff backing (PET film) was introduced to prevent elongation of ICE while peeling. Then ICEs were bonded onto substrates. The 90°-peeling tests were with a constant peeling speed of 50 mm min$^{-1}$ on a peel strength tester (GM-718).

**Decomposition voltage tests**. Decomposition voltage of ICE and hydrogel electrolyte were tested via Linear Sweep Voltammetry (LSV) on an electrochemical workstation (CHI660E), the samples were sandwiched by two steel round electrodes for tests, the scan rate was set at 0.5 mV s$^{-1}$ at the range from 0 to 10 V.

**Fabricating of the touch sensors**. The ICE (co-MPEG550A) was cut to a shape of $80.0 \times 15.0 \times 1.0$ mm$^3$. Two aluminum electrodes were then attached at both ends of the ICE for testing. Finger touched on the middle of the sensor without pressing for a touched measuring. The sensor was stretched to twice of the original length for a stretched measuring. The insulating film for covered touch sensor was silicone membrane.

**Data availability**. The data that support the findings of this study are available from the corresponding author upon reasonable request.

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

## Acknowledgements

This research was supported by the National Natural Science Foundation of China (Nos. 51773165), The Fundamental Research Funds for the Central Universities (xjj2015119) and Young Talent Support Plan of Xi'an Jiaotong University. We appreciate Mr. Junjie Zhang, Ms. Axin Lu (Instrument Analysis Center of Xi'an Jiaotong University), and Ms. Jingjing Liu for the valuable helps of testing.

## Author contributions

L.S. and S.D. conceived the idea and designed the research. L.S., T.Z. performed the experiments. G.G., X.Z., W.W., W.L., S.D. analyzed and interpreted the results. L.S. and S.D. drafted the manuscript, and all authors contributed to the writing of the manuscript.

## Additional information

**Competing interests:** The authors declare no competing interests.

