## [Peer Review File · Nature Communications]

Reviewers' comments:

Reviewer #1 (Remarks to the Author):

In this manuscript, the authors report the fabrication of a series of high stable ionic elastomers by introducing a lithium salt into the polymer matrix via photopolymerization. Meanwhile, the application of the elastomers as touch sensors has been demonstrated. The materials and fabrication procedures are simple but the mechanical/physical/electrochemical properties are highly interesting. This work would be highly valuable to the polymer community and might be extended to find other applications such as electronic skins and membranes in lithium-ion batteries. Therefore, the reviewer supports the publication of this manuscript in Nature Communications.

Some detailed comments that may be useful for authors are as follows:

1. The title of the manuscript could not deliver/highlight the novelty or importance of this work. Ionic elastomers are a very broad definition (see Structure and Properties of Ionomers, pp279.). Essentially all kinds of ionic elastomers could transport ions. Therefore, an appropriate title is required here.
2. "The tensile tests were performed on an electronic tensile machine with a 5000-N load cell". Considering that the fracture strength of the soft elastomers is only 0.35 MPa, a smaller load cell (<100 N) should be used to obtain accurate mechanical properties.
3. In Figure 2, the compositions of the hydrogel were not mentioned. Meanwhile, the origins of the strong adhesion between elastomer/metal and elastomer/elastomer should be
4. The ionic conductivities of the elastomers seem to be not astonishing ($\sim 10^{-7}$ S/cm at room temperature for PBA). Copolymerization with other monomers could lead to much increased conductivity (~ 100 -fold). Authors should summarize previous ionic elastomers with ionic conductivities and include the results in the supporting information followed by reasonable comparisons. Meanwhile, the mechanism that causes the "greatly enhanced conductivity via copolymerization" should be discussed and additional experiments/results should be provided. Moreover, how the copolymerization could affect the mechanical properties?
5. The ionic elastomers were used as touch sensors and the impedance spectra were used to distinguish the different stimuli. However, measuring the variations in impedance typically requires long acquisition times and the response time is way too long. Thus, this method may be not suitable for further practical applications. Developing capacitive force sensors using these elastomers would be more appropriate here.
6. An additional suggestion is that authors should wear gloves while doing all kinds of tests. The supplementary videos should be replaced with new ones.

Reviewer #2 (Remarks to the Author):

In this paper entitled "Ionic conducting elastomers", L. Shi, et. al describe a new type of conductor by embedding salt into elastomers. The elastomer is stretchable, transparent, and conductive (although with a low conductivity). The elastomer remains stable in air, in temperature above 100 °C, under voltage up to 10 V, and does not corrode metals. At the end, a touch sensor is demonstrated. But what are the differences between the ICE in this work and ionic liquid gels that also consist of polymer networks and ions. In addition, ionic liquids are stable in air, in high temperature, and have conductivity several orders of magnitude higher than the ICE.

Some other comments:

1. The text should be carefully polished. The tense should keep constant. Draft has several grammatical errors/ spelling mistakes. For eg:
 - a: on Page 3, ...photo-initiator 184 was dissolved, "was" should be "were";
 - b: on Page 4, Figure 1c, ICE possess good stretchability, "possess" should be "possesses";
 - c: on Page 5, ...frequency need to be well considered. "need" should be "needs";
 - d: on Page 6, ...Photographs in Figure 2C was ICE-metal and hydrogel-metal..., "was" should be "were".

And more.

2. Adhesive property of ICE should be characterized with standard tests and data should be shown if the authors emphasize the adhesion.

3. The first paragraph reviews progresses in conductive elastomers. Stretchable, transparent, conductive elastomers have been extensively studied. What are the advantages of the ICE in this work compared to existing conductive elastomer? What's the challenge this work resolve?

4. In Fig. 2E, the current changes negligibly over the test voltage range. It might be due to the large resistance of the ICE.

5. Experimental results are simply described without explaining and discussion, for eg: Why "thinner cover resulted in easier circuit change when finger touched"? How do the results guide the applications, uncovered sensor is better or sensor with thinner cover is better?

Responds to reviewers comments:

Reviewer #1:

In this manuscript, the authors report the fabrication of a series of high stable ionic elastomers by introducing a lithium salt into the polymer matrix via photopolymerization. Meanwhile, the application of the elastomers as touch sensors has been demonstrated. The materials and fabrication procedures are simple but the mechanical/physical/electrochemical properties are highly interesting. This work would be highly valuable to the polymer community and might be extended to find other applications such as electronic skins and membranes in lithium-ion batteries. Therefore, the reviewer supports the publication of this manuscript in Nature Communications.

Some detailed comments that may be useful for authors are as follows:

1. The title of the manuscript could not deliver/highlight the novelty or importance of this work. Ionic elastomers are a very broad definition (see Structure and Properties of Ionomers, pp279.). Essentially all kinds of ionic elastomers could transport ions. Therefore, an appropriate title is required here.

Responds: As the Reviewer's good advice, we have changed the title of the manuscript to "Highly stretchable, transparent ionic conducting elastomers".

2. "The tensile tests were performed on an electronic tensile machine with a 5000-N load cell". Considering that the fracture strength of the soft elastomers is only 0.35 MPa, a smaller load cell (<100 N) should be used to obtain accurate mechanical properties.

Responds: It is our negligence and we are sorry about this. According to comment, related content have been improved. We have re-performed the tensile test of the material with a smaller load cell (50N), and obtained more accurate mechanical properties. We have now replaced Figure 1C with the new version.

3. In Figure 2, the compositions of the hydrogel were not mentioned. Meanwhile, the origins of the strong adhesion between elastomer/metal and elastomer/elastomer should be

Responds: It is our negligence and we are sorry about this. The hydrogel for

comparison experiments was polyacrylamide(PAAm) hydrogel containing 2M lithium chloride (LiCl) salts. The hydrogel was synthesized via the method in Ref 20. We now mentioned the hydrogel in the main text, Figure 2 and methods. The strong adhesion property of the elastomer originated from the polymer substrate PBA, a commonly used Acrylate Adhesive. Meanwhile, Adhesive property of ICE have been characterized and data were shown in Figure 2.

4. The ionic conductivities of the elastomers seem to be not astonishing (~10⁻⁷ S/cm at room temperature for PBA). Copolymerization with other monomers could lead to much increased conductivity (~100-fold). Authors should summarize previous ionic elastomers with ionic conductivities and include the results in the supporting information followed by reasonable comparisons. Meanwhile, the mechanism that causes the “greatly enhanced conductivity via copolymerization” should be discussed and additional experiments/results should be provided. Moreover, how the copolymerization could affect the mechanical properties?

Responds: As the Reviewer's good advice, we have provided additional experiments/results. As mentioned in the main text, the ICEs was a kind of new material, which showed highly stretchability, high transparency and ionic conductivity at the same time. Meanwhile, homopolymer of the comonomers are frequently used solid-state polymer electrolytes. Ions can transport through the polymer backbones more freely. We showed that as the volume percentage of MPEG550A increases, the conductivity of the copolymer increases, and the homopolymer of PMPEG550A with 0.5 M LiTFSI have a ionic conductivity of 6.3×10^{-5} S/cm (Figure S5, S6). By introducing PMPEG550A into the polymer substrate helps increasing ionic conductivity. PMPEG550A solid electrolyte is brittle and unstretchable, in this study, copolymerization achieves high conductivity and stretchability.

5. The ionic elastomers were used as touch sensors and the impedance spectra were used to distinguish the different stimuli. However, measuring the variations in impedance typically requires long acquisition times and the response time is way too long. Thus, this method may be not suitable for further practical applications. Developing capacitive force sensors using these elastomers would be more

appropriate here.

Responds: Thanks for the Reviewer's advice. In this manuscript, we use impedance spectra and impedance complex plane to clarify different stimulus of the touch sensors. When getting a whole frequency range (1M Hz to 0.01 HZ) impedance spectra, it takes tens of minutes. But, when we fix a single frequency, as shown in Figure 4A-e ($f = 20$ KHz) and B-e ($f = 10$ KHz), we can accomplish real-time monitoring, getting both Z' and Z'' in a single acquisition. In this way, different stimuli will appear in different region in the two-dimensional impedance complex plane. Actually, the whole frequency range (1M Hz to 0.01 HZ) impedance spectra was to find an optimal frequency which will better distinguish the different stimuli, as shown in Figure 4A-d and B-d, then, we fix the single optimal frequency to detect the different stimuli, as shown in Figure 4A-e ($f = 20$ KHz) and B-e ($f = 10$ KHz).

6. An additional suggestion is that authors should wear gloves while doing all kinds of tests. The supplementary videos should be replaced with new ones.

Responds: Thanks for the Reviewer's kind suggestion, we will take care of our health more seriously when doing chemical related experiments. The supplementary video have been replaced by new ones. In the new version demonstrating adhesive property of the ICEs, we wear gloves doing the test.

Special thanks for your good comments.

Reviewer #2 (Remarks to the Author):

In this paper entitled "Ionic conducting elastomers", L. Shi, et. al describe a new type of conductor by embedding salt into elastomers. The elastomer is stretchable, transparent, and conductive (although with a low conductivity). The elastomer remains stable in air, in temperature above 100 oC, under voltage up to 10 V, and does not corrode metals. At the end, a touch sensor is demonstrated. But what are the differences between the ICE in this work and ionic liquid gels that are also consist of polymer networks and ions. In addition, ionic liquids are stable in air, in high temperature, and have conductivity several orders of magnitude higher than the ICE.

Responds: Thanks for the Reviewer's comments. ICEs are quite different from ionic liquid gels: a) Elastomers and gels are not the same thing. ICEs are elastomers with a

small amount of solid electrolyte salts with salt percentage <15 wt% and showed as elastomers, ionic gels contain a large amount of liquid electrolyte salts (ionic liquid) with salt percentage commonly >80 wt% and showed like gels. b) In the microscale, ions transport through polymer backbone in ICEs, while in ionic gels, ions transport through the liquid network, unionized ionic liquids in the polymer network serve as solvents. In addition, ionic liquid gels are not stable in air. Because the large amount of ionic liquid contained in the polymer network are easy to absorb moisture from air, especially in high humidity environment, though they do not evaporate as hydrogels. Then, the ionic gel will have disadvantages of hydrogels, such as corrosive to metals, low decomposition voltage.

Some other comments:

1. The text should be carefully polished. The tense should keep constant. Draft has several grammatical errors/ spelling mistakes. For eg:

a: on Page 3, ...photo-initiator 184 was dissolved, “was” should be “were”;

b: on Page 4, Figure 1c, ICE possess good stretchability, “possess” should be “possesses”;

c: on Page 5, ...frequency need to be well considered. “need” should be “needs”;

d: on Page 6, ...Photographs in Figure 2C was ICE-metal and hydrogel-metal..., “was” should be “were”.

And more.

Responds: Thanks for the Reviewer's advice. It is our negligence and we are sorry about this. We have polished the manuscript carefully.

2. Adhesive property of ICE should be characterized with standard tests and data should be shown if the authors emphasize the adhesion.

Responds: Thanks for the Reviewer's advice. Adhesive property of ICE have been characterized and data were shown in Figure 2.

3. The first paragraph reviews progresses in conductive elastomers. Stretchable, transparent, conductive elastomers have been extensively studied. What are the advantages of the ICE in this work compared to existing conductive elastomer? What's the challenge this work resolve?

Responds: Thanks for the Reviewer's comments. We have made revision of the current introduction form of the manuscript, answering the questions more clearly. ICEs are new materials, different from conventional conductive elastomers. ICEs are ionic conductors with very high stretchability and transparency, and the manufacturing process are quite simple and mild.

4. In Fig. 2E, the current changes negligibly over the test voltage range. It might be due to the large resistance of the ICE.

Responds: Yes. Large resistance helps highering the decomposition voltage. When a high voltage was applied between ICE, electrolytic reaction did not happen. The property enables high voltage or low current applications, such as lithium batteries and sensors. Moreover, because of the large resistance of ICE, it helps slow down the corrosion of metal-ICE junction, the electrolyte provide low ion transport and low current in corrosive microcells.

5. Experimental results are simply described without explaining and discussion, for eg: Why “thinner cover resulted in easier circuit change when finger touched”? How do the results guide the applications, uncovered sensor is better or sensor with thinner cover is better?

Responds: As the Reviewer's good advice, we have discussed our experiments in more detail. In addition, our sensor is a kind of surface capacitive touch sensor. When the cover is thinner, a bigger stray capacitance introduced in the circuit. Uncovered sensor introduce an even bigger stray capacitance, but uncovered sensor is unsuitable for further practical applications.

Special thanks for your good comments.

REVIEWERS' COMMENTS:

Reviewer #1 (Remarks to the Author):

In this revised manuscript, authors have addressed most of my concerns.

Reviewer #2 (Remarks to the Author):

The authors have carefully addressed the review comments. The reviewer would recommend the acceptance of this manuscript for publication in Nature Communications with the following issues being addressed:

1. The experimental section should be revised accordingly. For example, the 500-N load cell should be updated with 50-N load cell.
2. In Fig. 4, the applied stretch should be indicated. How does the measured signal vary with stretch? Will the touch position also affect the signal? Given a signal, how to tell whether it is a stretch or a touch?

Responses to Reviewers' comments:

Reviewer #1 (Remarks to the Author):

In this revised manuscript, authors have addressed most of my concerns.

Response: Special thanks for your good comments.

Reviewer #2 (Remarks to the Author):

The authors have carefully addressed the review comments. The reviewer would recommend the acceptance of this manuscript for publication in Nature Communications with the following issues being addressed:

1. The experimental section should be revised accordingly. For example, the 500-N load cell should be updated with 50-N load cell.

Response: We have revised the experimental section accordingly.

2. In Fig. 4, the applied stretch should be indicated. How does the measured signal vary with stretch? Will the touch position also affect the signal? Given a signal, how to tell whether it is a stretch or a touch?

Response: It is our negligence and we are sorry about this. The sensor was stretched to twice of the original length for a stretched measuring as schemed in Fig. 4a. We have now indicated the point in Method section.

Supplementary movie 2 demonstrates the measured signal vary with stretch at a fixed frequency: the impedance magnitude $|Z|$ increases as the increase of stretch, while the phase angle (ϕ) remain almost unchanged.

Touch position affects the signal, as shown in Fig.4e and Fig.4f, in the impedance complex plane, repeatedly detected data spread in corresponding regions. This originates from the slight difference of the touched position and the difference of contact area. In the main text, we have studied the stretch and touch of the sensor without varying the stretch and touch position, lest the data get too complicated. We have demonstrate them in Supplementary movies. We have revised our

Supplementary movie 3 to reveal the affection of touch position.

Given an impedance signal, as the signal is a complex quantity which can be expressed as (Z', Z'') or $(|Z|, \phi)$, it can be dotted in the impedance complex plane. Because different regions in the impedance complex plane correspond to different kinds of stimuli one-to-one (Fig.4e), then we can find which region of the given signal is in and tell what kind of the signal is, a stretch, a touch, or a stretch and touch.

Special thanks for your good comments.